# A hybrid human recognition framework using machine learning and deep neural networks

**Abdullah M. Sheneamer** [ID] ☯¤*, **Malik H. Halawi** ☯, **Meshari H. Al-Qahtani** ☯

Computer Science Department, Jazan University, Jazan, KSA

☯ These authors contributed equally to this work.
¤ Current address: Computer Science Department, Jazan University, Jazan, KSA
* asheneamer@jazanu.edu.sa

**Data Availability Statement:** I used published dataset after I got a permission from the author. Face Dataset: https://www.kaggle.com/datasets/vijaykumar1799/face-mask-detection Human Gait Dataset: http://www.cbsr.ia.ac.cn/english/Gait%20Databases.asp.

## Abstract

Faces are a crucial environmental trigger. They communicate information about several key features, including identity. However, the 2019 coronavirus pandemic (COVID-19) significantly affected how we process faces. To prevent viral spread, many governments ordered citizens to wear masks in public. In this research, we focus on identifying individuals from images or videos by comparing facial features, identifying a person's biometrics, and reducing the weaknesses of person recognition technology, for example when a person does not look directly at the camera, the lighting is poor, or the person has effectively covered their face. Consequently, we propose a hybrid approach of detecting either a person with or without a mask, a person who covers large parts of their face, and a person based on their gait via deep and machine learning algorithms. The experimental results are excellent compared to the current face and gait detectors. We achieved success of between 97% and 100% in the detection of face and gait based on F1 score, precision, and recall. Compared to the baseline CNN system, our approach achieves extremely high recognition accuracy.

## 1 Introduction

Globally, coronavirus illness (COVID-19) has had a substantial impact on people's day-to-day lives. Governments in different countries faced problems in the identification of people with the spread of COVID-19 [1, 2]. When people start wearing masks, it is difficult to discover the identity of the person because masks reduce the identification of a person. Some research has shown that the mask reduces the accuracy of finding the person [3, 4]. The current research study focuses on how masks affect how quickly these and other significant social aspects are processed. The masks are affected to find gender, age, and emotion [3–5].

Although the latest study broadens the dataset feature scope to include timed measurements and added major facial parameters such as gender and age. When leaving the house or heading to work, the World Health Organization has encouraged individuals all around the world to wear mask https://www.who.int/news-room/questions-and-answers/item/coronavirus-disease-covid-19-masks. It is difficult to check people manually in public places. Face masks also render older face recognition technologies ineffective because they are designed primarily for uncovered faces. This makes it necessary right to create a powerful

**Funding:** The author extends their appreciation to the Deputyship for Research & Innovation, Ministry of Education in Saudi Arabia for funding this research work through the project number ISP-202.

**Competing interests:** The authors have declared that no competing interests exist.

system that can locate persons wearing masks and identify different people wearing masks based on their walking patterns. The technology can identify masked faces and determine gait patterns using deep learning and machine learning techniques. Moreover, it is currently not possible to assess both face mask identification and gait type recognition using a single heterogeneous dataset. Facial recognition has become a worldwide study topic in recent decades. Furthermore, technical developments and the rapid growth of artificial intelligence have resulted in great progress.

In order to locate and control entry to airports, schools, businesses, and other sites, both public and private enterprises employ facial recognition systems. On the other hand, government bodies have adopted many biosafety legislations in response to the spread of the COVID-19 pandemic. Face masks should be needed in public areas since they effectively protect users and those around them. As the virus spreads by physical contact, traditional identification technologies (such fingerprints) and password entry on a keyboard become vulnerable [6, 7]. Therefore, facial recognition technologies are the perfect replacement since they don't require direct physical contact like other systems. The use of the face mask in these systems, however, has proven to be a substantial AI challenge.

When facial recognition is performed with only half of the face visible, numerous crucial pieces of information are lost [8]. This denotes the need to create algorithms that recognize a person wearing a face mask by finding the way of walking. Furthermore, it is critical to investigate ways to increase the masked facial recognition system's effectiveness during a pandemic. The hidden regions of the target face must be known before using the subtype of hidden face recognition known as "masked facial recognition".

The computer vision community is currently interested in a research area called occluded face recognition. Hidden face recognition systems have up to this point concentrated on discovering and recognising faces in the wild, even when the hidden parts of the face are in various positions and shapes. Meanwhile, a masked person's facial features, mouth, and cheeks, etc. are often hidden. The only features that are still visible may just be the forehead, eyes, and brows. Masked face recognition systems can therefore focus on analyzing details such as gait patterns. Face recognition is recognized as a real-world solved problem with near 100% recognition accuracy with the rapid growth of deep learning methods in computer vision [9].

To propose face recognition algorithms in particular domains, methods for evaluating textual and form features that are employed for facial recognition are currently being investigated. These feature types demonstrate their speedy processing capabilities without the need for a large amount of training data. Our research study provides a deep facial recognition model for masked faces combining feature extraction with deep learning using convolution neural network (CNN) architectures.

As a result, acting in situations where faces are only partially visible poses significant difficulties for typical human performance. Education, caregiver-patient relationships, and lipreading assistance for those with hearing loss are a few examples, but not all of them. When individuals wear masks, all these possibilities are hampered.

In this study, we will try to focus on how to discover the identity of the person if, for example, said person is wearing a mask, sunglasses, and a hat; the features of this person's face would be much lower if non-existence for we need more information on any person who covers his facial feature so we choose to add gait information thus we raised the chance of recognizing who this person is.

On the other hand, when gait is used alone is not enough, for example, if someone is wearing unusual clothes, clothes which cover their legs and feet, or if they are sickly, their gait might differ. Knowing that each person has a different gait or something special in the way they walk, and it is difficult for the person to change the way he walks, we can raise the

percentage, identify the person significantly, and we can find the wanted criminal or person quickly and efficiently.

It is important under the current circumstances and in the future, if a pandemic such as the coronavirus occurs and people are forced to wear the mask that there is a highly efficient technology that accurately identifies the person and there are a lot of sensitive places where this technology should be applied such as airports, banks, shops, government places and a lot, there are a lot of facial recognition systems but they are expensive and need a lot of time to evaluate.

There is also a problem of finding a person by fingerprint because they are vulnerable to viruses. Therefore, we developed this system to help effectively find the person without potential risk. The contributions of the paper are as follows.

- We aim to build a full study on how to detect people who wear masks, people who do not wear masks, and people who cover a large part of their face, such as wearing a hat, glasses, and mask, and finding them by walking.

- We also aim to create a comparison between deep learning and machine learning and find which is better to find people.

- We aim to build our technology at the lowest cost and at a speed to get to know people to be highly efficient for use

The rest of the paper is organized as follows. Section 2 discusses related work. Our proposed framework is introduced in Section 3. In Section 4, the framework's experiments are described. Threat of validity is covered in Section 5. Finally, the paper is concluded in Section 6.

## 2 Related works

### 2.1 Convolution neural network

Deep learning algorithms consistently outperform standard image categorization networks, especially when the CNN algorithm is used. Recent research has been performed on tuning CNNs for higher accuracy, whilst introducing several powerful CNN designs (such as AlexNet, VGGNet, GoogleNet, etc.).

There have been two recent changes in how to tune a CNN: The representation performance of deeper or broader architectures (such as DenseNet, Xception and Inception-v4) can be improved by increasing the number of trainable parameters, while other research has focused on developing small and effective CNNs because of their limited processing capacity (e.g., ShuffleNet, SqueezeNet and MobileNet-v2 etc.).

All these network architectures outperform conventional machine learning techniques; for example, support vector machines (SVM) based on oriented gradient histograms (HOG) and K-nearest neighbours (kNN), in classification tasks using the ImageNet classification dataset or the CIFAR-10 classification dataset. With the increasing depth and breadth of CNNs, overfitting issues have become apparent because of the limitations of the datasets and the effect on generalization of the network.

Altering neural network design by adding dropout layers is one way to prevent overfitting. Specific investigations were conducted into hyper-parameters in the selection of the training and regularisation conditions. To prevent overfitting, data augmentations such as random rotation, random trimming, and random mirroring are widely used [10].

### 2.2 DNN-based real-time face mask detection system

During the COVID-19 outbreak, there were major operational challenges for masked face identification and face mask detection [11]. Consequently, recent research initiatives have

focused on these two areas. Current research into face mask identification falls into 3 categories: classical machine learning methods (ML), deep learning-based methods, and hybrid approaches. Examples of hybrid approaches include algorithms combining traditional machine learning techniques with deep learning.

Deep learning-based methods are heavily involved in facial mask recognition, while conventional ML-based methods are limited in their use. Adrian et al. [12] noted that when medical staff in the operating room are not using face masks, an automatic mechanism sounds an alarm. For face detection, this system used the Viola and Jones face detector, and the Gentle AdaBoost for face mask detection.

Vijitkunsawat and Chantngarm [13] determined that the best model for face mask identification was by comparing two conventional ML classifiers, KNN and SVM, plus a DL technique, MobileNet. The outcomes revealed that, in terms of performance, Mobile Net outperformed KNN and SVM. The deep learning mode InceptionV3 [14] was used for automating the mask recognition process. The learned InceptionV3 model was adjusted to identify the images as masked or unmasked.

To train and evaluate the InceptionV3 model, the Simulated Masked Face dataset (SMFD) was used. Face mask identification was performed using the Deep Learning-based SSDMNV2 model. To classify the masked and unmasked images, the SSDMNV2 model uses a MobilenetV2 model with a face detection single-shot multi-box detector.

Ali et al. [15] researched face recognition in a video stream, using the Local Binary Pattern (LBP) histogram with processed data. This involves face detection using both Haar cascade files, incorporating nose, eye, and skin detection. The detected faces then act as the input for LBP, to improve the recognition system's accuracy. The system also facilitates creating a dataset comprising faces and corresponding names, which is used in the facial recognition phase. Experimental results of the proposed system achieved 96.5% recognition accuracy.

Ali et al. [16] employed an enhanced approach to improving face detection by combining Haar cascade files with nose, eye, and skin detection methods. The suggested system is evaluated using the Wild database, and the results reveal a detection accuracy of up to 96%. Comparison with other face detection systems reinforces the effectiveness of the suggested method. For face mask identification, a real-time end-to-end network based on VGG-16 was demonstrated. For masked facial detection, a cascaded CNN system with three CNN models was proposed. Five layers comprise the initial CNN model, whereas seven layers each were used in the second and third models. The negative aspect of the three cascaded CNNs is that they require more computing power. Current research has employed hybrid models that combine ML and DL-based methodologies in addition to standalone deep learning models.

Based on conventional ML and DL-based techniques, Bhattacharya et al. [17] developed a hybrid model for face mask recognition. This hybrid model trains an SVM, a combined algorithm, and a decision tree to classify images using the Resnet 50 feature extraction algorithm into masks and non-masks.

A Hybrid-Face-Mask-Net model was suggested for recognising face masks. The main components include deep learning, an original feature extractor, and traditional machine learning classifiers. Deeper learning (CNN) and manual feature extraction approaches were used for extracting substantial characteristics because of the lack of data. Feature selection was then conducted. Finally, a random forest classifier was used to perform Nagrath's classification [18].

Eman et al. [19] introduced an innovative approach for masked face recognition, leveraging a multi-aspect methodology combining deep learning-based mask detection using pre-trained ssdMobileNetV2, landmark and oval face detection to identify crucial facial features, and robust principal component analysis (RPCA) to distinguish occluded and non-occluded

components in facial images. To enhance performance, particle swarm optimization (PSO) is employed to optimize both KNN features and the number of neighbors (k) for K-nearest neighbors (KNN). Experimental results highlight the superiority of the proposed method, demonstrating heightened accuracy and robustness to occlusion compared to existing approaches. Notably, the proposed method achieves an impressive recognition rate of 97%.

## 2.3 Non-locked face detection

Face recognition is a technique for estimating the bounding box of a person's face in a photograph or video image as shown in Fig 1. Face recognition in ordinary situations where people's faces are visible in the photos is called non-obscured face recognition.

Face detection, facial features recognition, facial features extraction, and facial features classification are the 4 basic processes of a standard face recognition algorithm. They should be detected when there is another face in the photo. Face detection is one of the first stages in the face recognition pipeline and one of the most important. Face recognition algorithms must

(a) 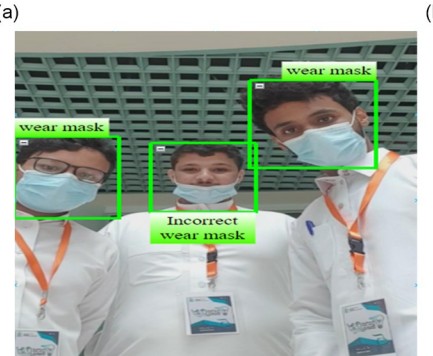 (b) 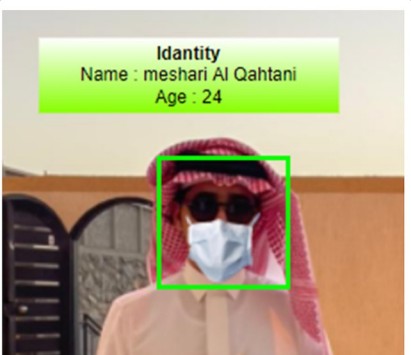 (c) 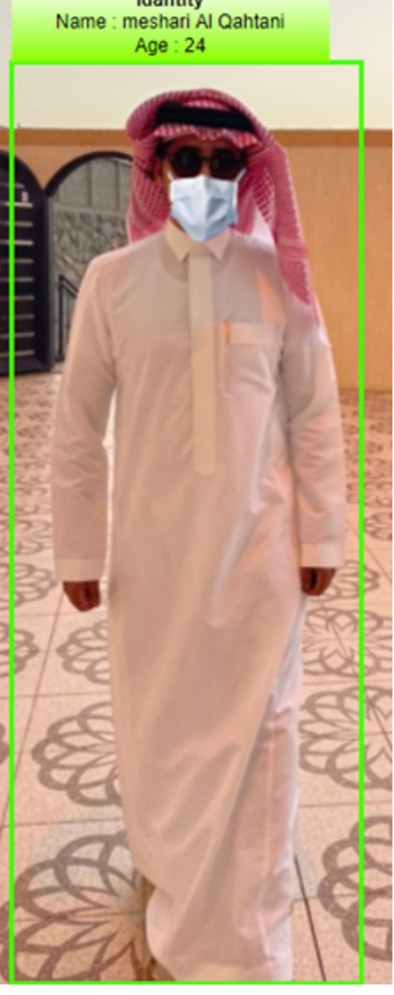

**Fig 1. Cases of recognition.** (**a**) Mask or with out mask. (**b**) Cover Large parts of their Face. (**c**) Walking.

therefore be resistant to positional illumination variations and scale discrepancies to supply strong face recognition performance.

Background noise should be minimised as far as possible [20]. In the literature, various facial recognition techniques have been reported. Viola and Jones [21] proposed the use of real-time hair-like features for frontal faces. Conversely, Viola-Jones face recognition, is not resistant to adjustments in position, lighting, or occlusion.

Additionally, colour characteristics have been used to identify faces in pictures [22]. Recently, Deep Learning has improved object recognition, resulting in successful Deep Learning face detectors. Many practical applications use the multitask cascaded convolution network (MTCN) face detector framework. To predict landmark locations in the fine range, MTCN uses a cascaded architecture with three levels of deep convolutional networks. MTCN outperformed conventional face detectors by a wide margin in several benchmarks.

Qi et al. [23] addressed the issue of fixed input image sizes and the limited generalizability of previous techniques. Thus, we produced a process based on cascaded convolutional networks which can recognize human faces in real-time with accuracy comparable to ultramodern architectures. To extract the required facial features like the corners of the eyes, brows, lips, etc., face feature detection is crucial. These points are used as inputs for face alignment. Face recognition accuracy has been improved by aligning the face to a fixed perspective.

To evaluate the face feature recognition algorithm, Rathod et al. [24] used the ground truth-based localization error. The performance of facial feature recognition algorithms has dramatically increased with the introduction of deep learning techniques. The third stage of the face recognition process is the extraction of facial features. In the past, a person's face's geometrical, textural, and geometrical aspects were used to determine their facial features.

Ding et al. [25] identified facial expressions, and local binary pattern features were extracted from movies. By employing an eye recognition module to locate the centre and corners of the eye, they developed a model that could locate a specific person. Some classification techniques are utilised to get the final recognition results after extracting facial features in the work of Gumus et al. [26].

## 2.4 Recognizing people using gait

The distinctive characteristics of humans make them distinguishable from each other. An automated method of detecting a person based on psychological or behavioural traits is known as biometric recognition, or simply biometrics. The benefits include that, unlike passwords or PINs, the features under investigation are a portion of personal information that is challenging to fabricate, exchange, or forget. The ability of biometrics to be utilised for negative recognition is another advantage.

One of the distinguishing qualities that can identify someone is how they walk, Jain et al. [9]. Gait is a sophisticated spatio-temporal biometric that describes how a person walks. Studies in a variety of disciplines, including psychology, mathematics, and health, demonstrate that a person's gait is a distinctive and differentiating characteristic, demonstrating the viability of human gait identification.

Johansson made the first attempt to prove this in 1970 when he discovered that it was possible to recognise well-known people by their movements by watching film clips of people walking while having light bulbs inserted in certain body areas, like joints. The first notable benefit of gait recognition is that it can be performed at a distance, a distinct advantage over other biometrics like face or iris recognition.

Gait is hard to disguise because most people walk to or enter a location. However, an individual's privacy may be compromised because recognition might be performed without the

person's knowledge or consent, raising ethical considerations. Although gait recognition is simple in normal circumstances, even minor changes might have significant repercussions. For example, a change of clothes has an impact on a person's movement, and lighting changes might harm the system's effectiveness. The viewing angle is also crucial as effective matching necessitates that the subject moves in the direction that had already been saved into the database.

Since walking pace significantly affects the physical traits of the person being viewed, it is a decisive element in identification. Most recognition algorithms assume that a person's walking speed is constant and predictable by using the gait time as their primary statistic. A person's weight, limb length, shoe type, posture, and other factors can all affect their gait. Among other gait recognition measures, these factors influence the walking tempo, stride length, walking silhouette with time, and angle of the limb segments, Fazenda et al. [27].

## 2.5 Automated person identification system using walking pattern biometrics

In 1997, a team of researchers introduced the concept of "footstep" biometrics, revealing that everyone's footsteps have unique characteristics. This makes footsteps beneficial in identity verification, as they mirror human behaviour, making replication challenging for others. By using footstep biometrics, it is possible to differentiate between various walking patterns and even identify or locate individuals based on their gait. The use of footstep sensors in smart home environments adds a multitude of applications, enabling the system to determine a person's position within a room. Smart homes, designed for effective communication with their inhabitants, can recognize human behaviour and interactions. The use of walking pattern recognition through footstep biometrics allows for efficient person location without direct visibility or interaction. Security systems, aiming for inconspicuous identification mechanisms, often face vulnerabilities in existing biometric recognition systems. Footstep biometrics, however, presents a unique challenge to imitation due to its reflection of an individual's conduct. The speed of locating a person is notably enhanced with footstep biometrics compared to traditional methods requiring physical interaction, such as fingerprint or iris scans. Unlike conventional identification processes where individuals actively participate by placing their finger on a scanner or exposing their iris, footstep recognition operates subtly and without the person's explicit awareness of identity processing. This automated nature sets footstep biometrics apart from other systems, making it applicable in diverse settings, including army bases, smart homes, bank vaults, and various other locations. Furthermore, the potential extends to military applications, where a wireless sensor network on the battlefield could utilize footstep biometrics to track adversaries' movements. The study introduces an innovative approach to extracting information from a person's gait or body movements, proposing a system that leverages this data for more accurate, in-person identification, Rahman et al. [28].

## 2.6 Recent related work

Rani and Kumar [29] offers a comprehensive exploration of the techniques and methodologies utilized in this field, covering aspects such as the framework, historical context, and key parameters. The review extensively investigates various classifiers employed, encompassing both traditional methods and those based on deep learning. It also scrutinizes the diverse datasets utilized in experimental studies and outlines the methodology used to evaluate articles focused on gait recognition. Anticipating a broad spectrum of future applications, particularly in security contexts, the systematic review underscores the promising potential of gait recognition technology.

Sahu et al. [30] aims to provide a comprehensive understanding of the techniques employed in gait recognition, with the goal of enhancing accuracy across various gait databases. Gait recognition, emerging as a prominent behavioral biometric trait based on human walking styles, offers unique features for identifying individuals at a limited distance with minimal cooperation, distinguishing it from other biometric techniques like voice, hand, face, iris, and fingerprint. Despite factors such as viewing direction, clothing type, shoe type, walking surface, speed, and object carriage influencing the correct classification rate, it is crucial to assess these covariates to develop a robust and dependable gait recognition system. The fundamental components of a gait recognition system involve the acquisition of gait data, background subtraction, feature extraction, feature selection, and classification.

The identification of individuals through behavioral biometrics, particularly focusing on gait as a distinctive trait reflecting one's walking style, has garnered significant interest within the biometrics industry. Traditionally, gait recognition relied on handcrafted approaches, but these methods faced challenges due to the impact of various covariates. The advent of deep learning, an emerging algorithm in biometrics, has shown promise in addressing these covariates and delivering highly accurate results. Mogan et al. [31]offered a comprehensive overview of the current state of deep learning-based gait recognition approaches, highlighting their capabilities. Additionally, they provided 286 a summary of the performance of this approach across diverse gait datasets.

## 3 Proposed work

Our approach places a surveillance camera at an entrance; if he is not wearing a mask, the camera will take his face data and send it to the server to see if that person represents a danger or not. If he does represent a risk this will send an alert to the observer employee to warn him. If the person is also wearing a mask, the camera will take his information and send it to the server, if his information is not correct enough, it will take his walking information and consider whether he is a risk or not. Governments, based on their databases, can also detect criminals and suspects through public space cameras as well as store cameras if they have control over them. Deep learning-based convolution neural network (CNN) patterns are used to design an effective network for detecting face mask images. Our primary goal is to train a custom CNN method to find people who cover large parts of their faces. Our technique depends on the following: 1. Face Detection. 2. Gait Recognition.

- Data can be gathered using a video camera, and a person's gait characteristics can be decided by analysing the photos. According to research, a person's walking movement differs from that of another. Therefore, the gait action could be a useful method by which to record a person's individuality [32].

- The sound of a person walking can also be a pattern. According to the study, a person's walking sound can be used to find them precisely and the accuracy rate is approximately 66%. The gait sound, according to the study, also transmits gender information. Gender may be decided with greater than 77% accuracy by the gait sound.

- Another solution to the issue might be the sensor-based system. To gather data regarding footsteps, several sensors are available, including pressure sensors. That might be put on the ground or in the wearer's shoes. Additionally, the Ground Reaction Force (GRF) can provide accurate information on footfalls.

- Ubi Floor is a different user identification system that uses 144 inexpensive ON/OFF switch sensors to detect the user's location and identify them based on their movement patterns

[28]. If there are obstacles in finding individuals with masks, using gait to find individuals is considered the best solution in finding them. There are numerous techniques to get data about gait patterns. The following describes a few of them:

## 3.1 Face detection

The study used deep learning and machine learning to minimize parameters while trying to achieve competitive accuracy is the trade-off between accuracy and inference time. To significantly improve the evaluation and training of the detection model a multitask loss, a new loss function, has been suggested, which combines four parameters Four sub-losses make up the multitask loss, which must be reduced. For each incorrect face categorization prediction, the model is further penalised. The second method is known as facing box regression and measures the separation between the predicted face's bounding box coordinates and the ground truth connected to the positive anchor. Box regression loss is similar to face landmark regression. We determine the separation between his predicted 5 face landmarks and the labelled landmarks rather than utilizing bounding boxes. The disparity between the original face and the reconstruction produced by the mesh decoder is the final cause of the dense face regression loss from a mesh decoder. All loss-balancing parameters for the theoretically derived multitask loss function are set to 0.25, 0.1, and 0.01, correspondingly. A collection of five landmark points and a dense 3D point mapping are output by default by AI in addition to the face position for the reconstruction procedures. Moreover, while MTCNN offers strong detection performance, the prediction speed-up is insufficient for real-time processing without GPUs. The following section introduces ultramodern approaches to face recognition, Turkoglu and Arican [33].

1- **Feature Extraction**. In this step, high-resolution features will be extracted for each part of the face while keeping its accuracy. The data may be ridiculously large or complex, so we need to reduce it without damaging it or diminishing its accuracy. The types of extraction of features differ according to our data. The data is on the face. The extracted features will be the eye, mouth, nose, the shape of the face, and ear. Data extracted from walking will be the way to walk and measure each part that moves in a person's body, such as the movement of the arms, feet, and head as seen in Fig 2. 2- **Face Recognition**. To get to know, the person requires that we have a dataset to compare the target data and the data in the dataset. So here, each part of the features will be analysed and compared correctly. When discovering the highest match percentage, the information available in the dataset is taken and sent to the last part.

3- **Identification**. Here, all information related to the target person will be obtained.

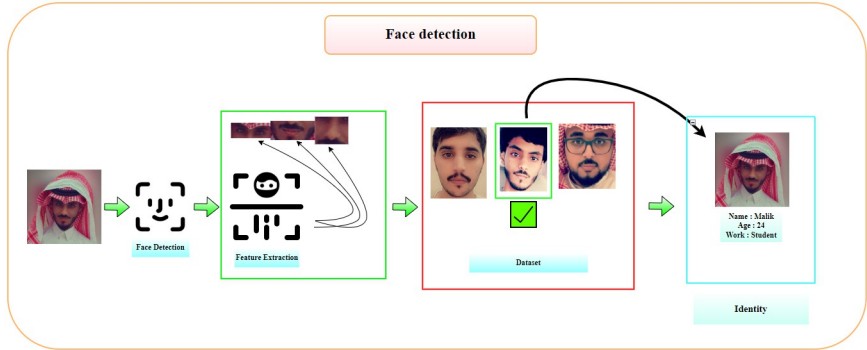

**Fig 2. Face detection work.**

### 3.2 Gait recognition

Gait analysis for human identification, which distinguishes the quality of a person, is recognition as seen in Fig 3. The term "biometric recognition" describes the automatic identification of people using feature vectors based on their physiological and behavioural traits. In many crucial applications, biometric technologies for remote human identification are growing in popularity. In these types of scenarios, gait recognition/identification becomes more appealing. Because each person has distinctive features, the term "biometrics" refers to one of such characteristics. Biological and behavioural traits make up biometric traits.

The face, fingerprints, iris, palm print, and DNA are physical traits. Voice and gait are other behavioural traits. Gait recognition is more appealing because these physiological parameters do not produce adequate results at low resolution and require user assistance. Finding a person using their gait entails observing how they move or walk. Low-resolution photos can also be used for gait recognition." According to the definition of gait, "a particular gait or manner of walking with foot." Recognition of human gait is based on the finding that each person's walking style is distinct and can be used to find them. Gait recognition systems can be categorised as appearance-based or model-based depending on how the features are extracted. Due to changes in viewing or movement directions, appearance-based techniques suffer from variations in appearance. Model-based techniques match their models to the input photos to extract the motion of the human body.

Model-based methods are viewed as scale-invariant. The technology will locate the unauthorised person and identify him by comparing his gait to sequences that have been previously stored. Gait recognition is typically done via background subtraction. Moving items are subtracted from the background using the background subtraction method to create a binary silhouette, Xu et al. [34].

A significant problem is the extraction of suitable features that correctly reflect the gait characteristics. Because it records the motions of most body parts and stores structural and transitional information, a person's silhouette is an interesting feature to study. It is unaffected by the wearer's clothing, and as it is simple to create from infrared imagery [2, 35], it supports night vision. The dimension of the observation vector decides how well the HMM may be trained. Due to its size, the silhouette information cannot be used. For better

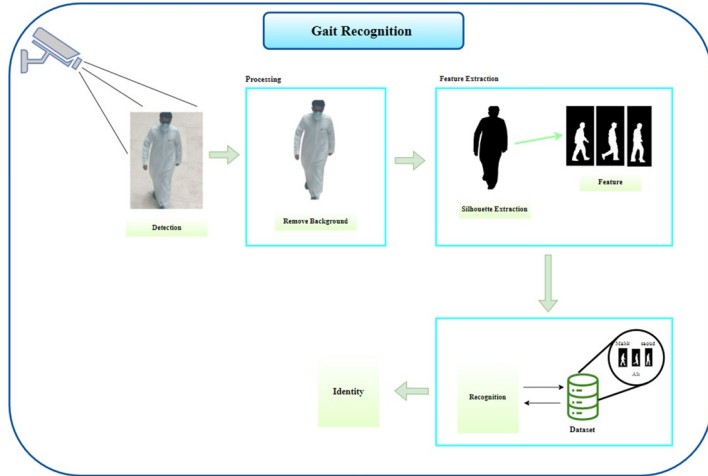

**Fig 3. Gait recognition work.**

performance, the information stored in the silhouette must be compressed. Now, we'll go over how to effectively encode this data. The training and evaluation module descriptions are then given in detail.

**1- Detection**. The person is first detected by the camera. The image is to be non-processed and there is a lot of information and dispersed objects which reduce the accuracy of the person's identification. The camera can decide the object in the dark areas by infrared.

**2- Processing**. At this point, all junk information is to be eliminated and all moving objects are found. The background is to be removed and focused only on the target to ease verification.

**3- Feature Extraction**. Extracting features in walking is to be different from extracting features in the face. Here, the movements of the body, such as the hand, feet, and head, are to be extracted and measured.

**4- Recognition**. Finally, the extracted walking data is to be compared with the walking data we have in the dataset and target recognition.

## 3.3 Algorithm

The camera is supposed to be stationary in the trials. Given a subject's image sequence, the silhouette is formed as follows:

1. Background subtraction is used to identify moving objects in each frame to reduce the effects of slight distortions in the background. The blob tracker then evaluates the fastest-moving objects in the scene [36].

2. To remove unwanted noise from the moving image, a typical 33 erosion filter is used.

3. Since we are only concerned with the body's exterior contours, we use weighted low-pass filters to estimate pixel intensities from the left and right margins of the picture to trace the body's left and right limits.

4. After that, the width of each row's silhouette is saved. The difference between the rightmost and leftmost boundary pixels in a row is all that determines its width. This work seeks to set up a system for automatically recognising walking patterns using spatiotemporal silhouette analysis. The gait refers to the dynamics and appearance of human walking action. The ability to intuitively recognise a person by gait depends on how the silhouette changes during the visual series. An overview of the proposed algorithm is shown in Fig 4. Human detection and tracking, feature extraction, and training and classification are its three key components.

The program's following segments locate and track the strolling figure in a sequence of pictures. To distinguish movement from the surrounding area and the moving area that corresponds to the spatial silhouette of the walker, a background estimation approach [37] is applied. A basic correspondence approach is used to follow Fig 4 sequentially. The second part takes the binary silhouette from each frame and transforms it into four 1D distance signals using four silhouette projections. The shape alterations of these silhouettes are converted into four sequences of four 1D distance signals to replicate temporal variations in the gait pattern. The estimation of the gait cycle, which is essential for gait identification, is likewise done using these one-dimensional signals. In the last stage, either principal part analysis (PCA) on the time-varying distance signals is used to determine the key elements of gait characteristics (training) or non-parametric pattern classification methods in the lower dimensional Eigenspace are employed to identify the person [38].

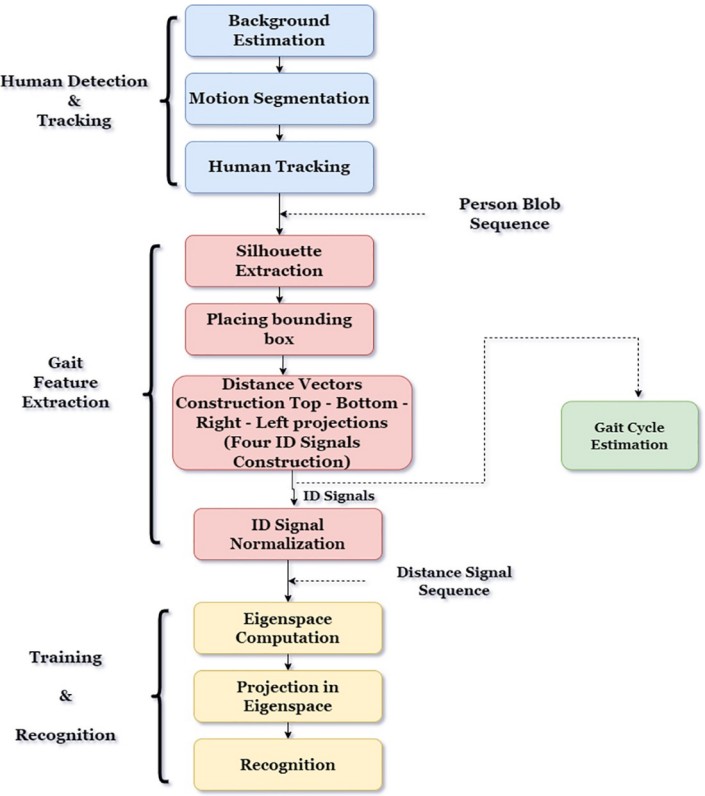

**Fig 4. Our proposed work.**

**Algorithm 1** Human Reconition based on Face and Gait Patterns

```
1: INPUT: F // Target Frames of Videos.
2: OUTPUT: HR //Classified Human Recognition not wearing a mask.
3: Steps:
4: for i = 1 ··· |F| do
5:   Visulaize the image in two categories (wearing mask or not).
6:   Convert RGB image into Gray-scale image.
7:   if F_i is wearing Mask then
8:     Identify moving objects.
9:     Remove unwanted noise from the moving image.
10:     Weighted low-pass filters.
11:     Silhouette extraction and placing bounding box.
12:   else if F_i is not wearing Mask then
13:     Resize the gray-scale into 100 × 100
14:     Normalize the image and convert it into 4 dimensional array.
15:   end if
16: end for
17: for building the CNN models based on face do
18:   Add a Convoluation layer of n filters.
19:   Add the second Convolution layer of n filters.
20:   Insert a Flatten layer the network classifier.
21:   Add a Dense layer of n neurons.
22:   Add the final Dens layer with n outputs for humans.
23: end for
24: for building the CNN models based on gait do
25:   Add a Convoluation layer of n filters.
```

```
26:    Add the second Convolution layer of n filters.
27:    Insert a Flatten layer the network classifier.
28:    Add a Dense layer of n neurons.
29:    Add the final Dens layer with n outputs for humans.
30: end for
31: Split the data, train the models and reconize the human
```

## 4 Evaluation/ Experiments

### 4.1 Dataset description

The research will be based on two types of databases with different images, backgrounds, and motions to cover all possibilities that might happen. In addition, these two types of databases will be merged and made dependent on each other. The first dataset on face detection https://www.kaggle.com/datasets/vijaykumar1799/face-mask-detection. It has 8,982 Images with different wallpapers, various positions, features, and different ethnicities. It consists of three classes with 2,994 Images for each class. The classes are those who wear a mask, do not wear a mask and those who wear a mask incorrectly as shown in Fig 5. The second dataset is Gait Recognition (CASIA-A) http://www.cbsr.ia.ac.cn/english/Gait%20Databases.asp. CASIA-A has 19,135 silhouettes with different walking positions ash shown in Fig 6. It consists of twenty classes. Each class in a different name and different images and has different walking corners Created by Wang et al. [39].

The dataset utilized in our Hybrid Human Recognition Framework includes CNN, VGG16, VGG19, AlexNet, and other deep neural network models, together with a wide range of features intended to enable thorough training and assessment of these models. The dataset includes a wide range of photos taken in different conditions to test the recognition framework's generalizability and robustness.

Variations in lighting situations, facial expressions, positions, and accessories like masks, glasses, and hats are among the dataset's detailed properties. The collection also contains occasions where individuals are not directly facing the camera, offering a realistic simulation of

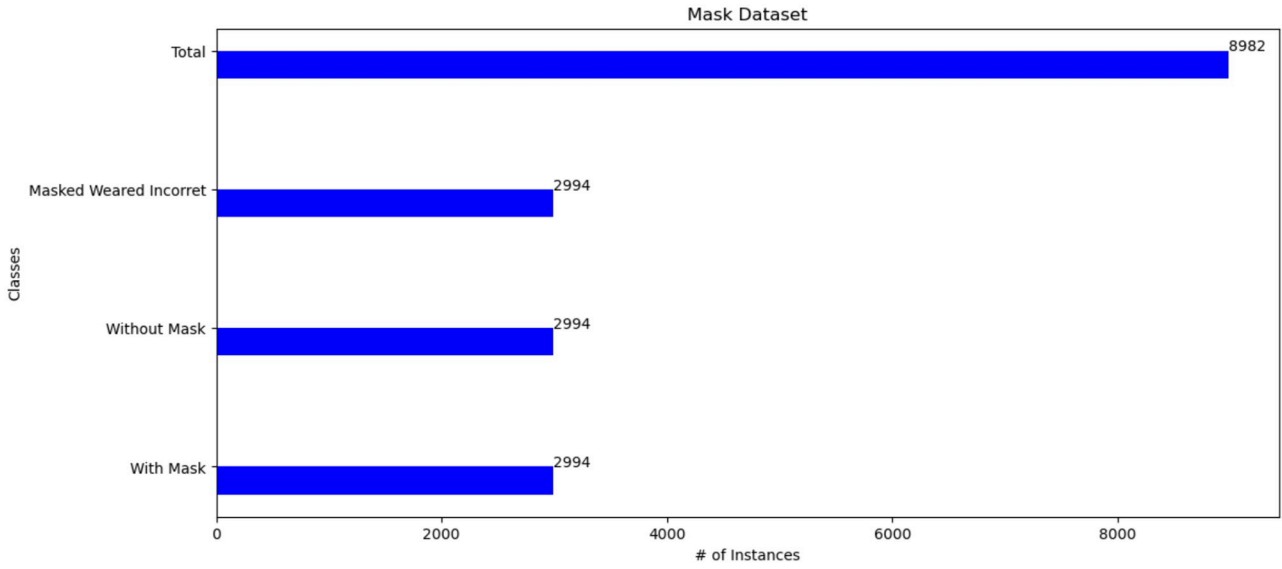

**Fig 5. Human face dataset description.**

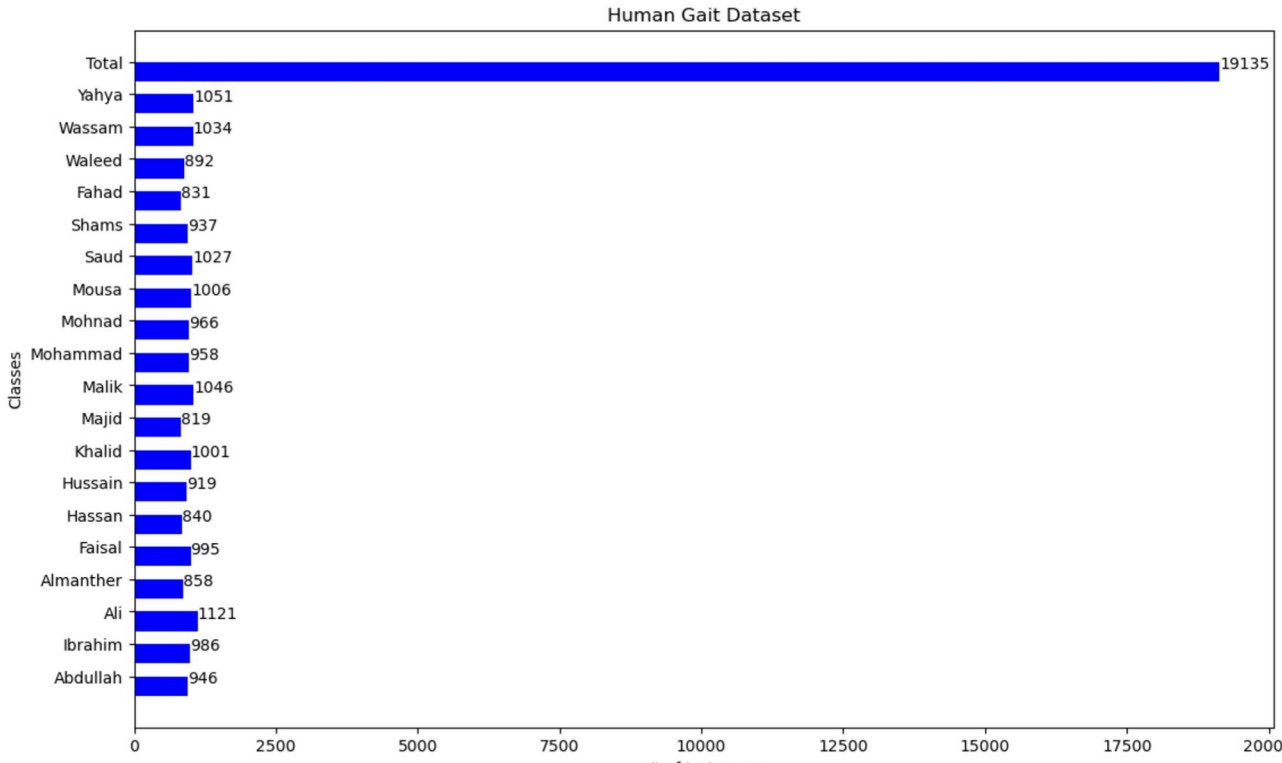

**Fig 6. Gait recognition (CASIA-A) dataset [39].**

real-world surveillance circumstances. In addition, to tackle current issues, the dataset has pictures of people wearing and not wearing masks, encompassing a variety of mask kinds and situations in which masks could be worn improperly. Because of the scale of the dataset, the models are exposed to a wide variety of environmental factors and human qualities. The collection might include, for instance, tens of thousands of high-resolution photos, each annotated with information about walking patterns, mask presence, and facial coverings.

The models, such as CNN, VGG16, VGG19, AlexNet, and others, can learn complex patterns and variances in human identification because of the dataset's extensive nature and variety of attributes. This guarantees that the Hybrid Human Recognition Framework is capable of handling real-world situations, which makes it a strong option for use in a range of sectors, such as public health, security, and surveillance.

## 4.2 Hyberparameters of CNN architectures and results

We generated comprehensive results for assessing the strength of our suggested model in the detection of human faces and gait. We experimented with several machines and deep learning algorithms to show that our approach can achieve high detection accuracy. We chose hyperparameter values for training a deep learning model can depend on several aspects, such as the specific architecture of the model, the nature of the data, and the problem being addressed. Hyperparameters are usually set through a combination of heuristics, empirical testing, and sometimes more advanced optimization techniques.

Hyperparameters as shown in Table 1, play a crucial role in the performance and training dynamics of Convolutional Neural Network (CNN) architectures. Hyperparameters of CNN

**Table 1. Hyberparameters of CNN architectures.**

| Parameter | Value |
| --- | --- |
| Batch size | Dataset size |
| Learning rate $\theta$ | 0.001 |
| Regularization srength $\beta$ | 0.001 |
| Input shape | 128 |
| epoch | 300 to 1000 |
| Activation Function | ReLU, softmax, and sigmoid |
| Momentum | 0.9 |
| Optimizer | SGD(lr = learning_rate, momentum = momentum) |
| Loss function | $categorical_crossentropy$ and $binary_crossentropy$ |

architectures, such as learning rate, batch size, regularization strength, activation function, optimizer, and number of epochs, significantly affect the model's training dynamics and performance. Careful selection and tuning of these hyperparameters are necessary to achieve the best results for a given task and dataset.

## 4.3 Machine and deep learning

Experiments were carried out in machine learning using face and gait images, using several algorithms and deep learning architectures. The baseline involves traditional machine learning algorithms, including Decision Tree (DT) [40], Random Forest (RF) [41], Support Vector Machine (SVM) [42], Naïve Bayes (NB) [43], and K-nearest Neighbors (KNN) [44]. The dataset was stratified, with 67% used for training and 33% for testing to ensure a fair representation of each class during model training. Four classifiers were used for building and comparing predictive models: RF, DT, NB, SVM, KNN, and deep learning architectures such as VGG16, VGG19, and AlexNet. Model performance was evaluated using four metrics: F1 score, recall, precision, and accuracy, which gauge the effectiveness of human recognition and performance predictions. F1 score results are presented in Tables 1 and 2. Fig 7a demonstrates that SVM achieved an overall accuracy of 91.26% for detecting faces and 97.92% for detecting human gait. In Figs 7b and 8, the AlexNet model achieved 99.28% accuracy for detecting human faces and 97.84% for detecting human gait. Notably, the deep learning classifiers, especially the AlexNet architecture, consistently outperformed traditional machine learning algorithms in all scenarios.

Figs 8–11 show the outcomes of the human face and gait detection utilising deep learning algorithms. The graphs illustrate the validity, accuracy, and loss of training for each of the three deep learning architectures. In terms of accuracy and loss of training and validation curve, the VGG19 model is the best deep-learning classifier for identifying human faces, as shown in Fig 10a. AlexNet is ranked first for detecting human gait as in Fig 8b, and the VGG16 model ranks second for detecting human face and gait regarding accuracy and loss of training and validation curve.

The results in face detection and gait recognition, comparing VGG19 with traditional CNNs as seen in Figs 10 and 11 respectively, demand a thorough analysis for meaningful insights. In face detection, VGG19's deeper architecture may be better able to capture intricate facial features, potentially leading to more accurate recognition. However, the computational cost and memory requirements should be critically assessed, especially in real-time applications. Traditional CNNs, with their lighter architectures, may excel in scenarios where computational efficiency is crucial. Conversely, gait recognition involves the study of walking

**Table 2. Performance assessments of different detection deep learning architectures for gait (CASIA-A) dataset.**

| Class | AlexNet | | | VGG16 | | | VGG19 | | | CNN | | |
|---|---|---|---|---|---|---|---|---|---|---|---|---|
| | Precision (%) | Recall (%) | F1-Score (%) | Precision (%) | Recall (%) | f1-Score (%) | Precision (%) | Recall (%) | F1-Score (%) | Precision (%) | Recall (%) | F1-Score (%) |
| Abdullah | 98.64 | 98.64 | 98.64 | 91.35 | 96.1 | 93.76 | 92.1 | 95.89 | 93.95 | 93.5 | 98.63 | 96 |
| Ibrahim | 100 | 100 | 100 | 100 | 97.36 | 98.66 | 100 | 100 | 100 | 100 | 98.61 | 99.3 |
| Ali | 97.7 | 97.7 | 97.7 | 96.73 | 98.88 | 97.8 | 91.3 | 98.82 | 94.91 | 92.77 | 89.53 | 91.12 |
| Almanther | 100 | 98.50 | 99.24 | 100 | 100 | 100 | 97.01 | 98.48 | 97.74 | 98.36 | 93.75 | 96 |
| Faisal | 95.31 | 96.82 | 96.06 | 94.28 | 95.65 | 94.96 | 93.84 | 91.04 | 92.42 | 87.5 | 91.3 | 89.36 |
| Hassan | 98.27 | 100 | 99.13 | 100 | 100 | 100 | 98.36 | 96.77 | 97.56 | 94.11 | 98.46 | 96.24 |
| Hussain | 97.26 | 100 | 98.61 | 98.63 | 98.63 | 98.63 | 100 | 95.71 | 97.81 | 97.18 | 94.52 | 95.83 |
| Khaled | 93.82 | 100 | 96.81 | 94.93 | 94.93 | 94.93 | 93.02 | 96.38 | 94.67 | 83.51 | 96.2 | 98.41 |
| Magid | 100 | 100 | 100 | 100 | 100 | 100 | 100 | 100 | 100 | 100 | 100 | 100 |
| Malik | 93.67 | 91.35 | 92.50 | 84.21 | 98.76 | 90.9 | 91.13 | 83.72 | 87.27 | 85.71 | 83.54 | 84.61 |
| Mohammed | 96.96 | 96.96 | 96.96 | 98.64 | 96.05 | 97.33 | 97.26 | 98.42 | 95.3 | 89.33 | 88.15 | 88.74 |
| Mohnad | 100 | 98.46 | 99.22 | 97.29 | 100 | 98.63 | 100 | 98.68 | 99.33 | 98.71 | 100 | 99.35 |
| Nasser | 100 | 100 | 100 | 100 | 100 | 100 | 100 | 100 | 100 | 100 | 100 | 100 |
| Saoud | 95.12 | 97.50 | 96.29 | 97.5 | 96.29 | 96.89 | 84.15 | 100 | 91.39 | 88.23 | 91.46 | 89.82 |
| Shams | 98.43 | 96.92 | 97.67 | 88.31 | 94.44 | 91.27 | 87.8 | 97.29 | 92.3 | 89.47 | 90.66 | 90.06 |
| Fahad | 94.82 | 91.66 | 93.22 | 88.7 | 82.08 | 96.49 | 98.18 | 84.37 | 90.75 | 80.64 | 78.12 | 79.36 |
| Waleed | 97.84 | 96.80 | 97.32 | 98.41 | 88.57 | 93.23 | 95.16 | 90.76 | 92.91 | 92.53 | 87.32 | 89.85 |
| Wassam | 100 | 93.45 | 96.61 | 98.66 | 94.87 | 96.73 | 100 | 97.5 | 98.73 | 93.67 | 89.15 | 91.35 |
| Yahya | 95.45 | 99.05 | 97.22 | 95 | 91.56 | 93.25 | 96.1 | 88.09 | 91.92 | 90.47 | 91.56 | 91.01 |
| Mousa | 100 | 100 | 100 | 97.5 | 97.5 | 97.5 | 98.68 | 98.68 | 98.68 | 98.7 | 91.56 | 95 |

(a)

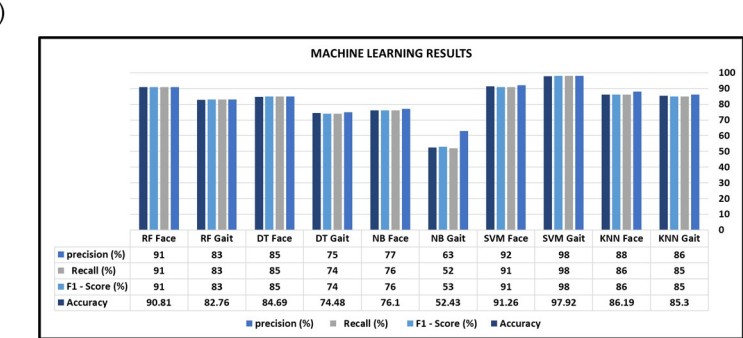

(b)

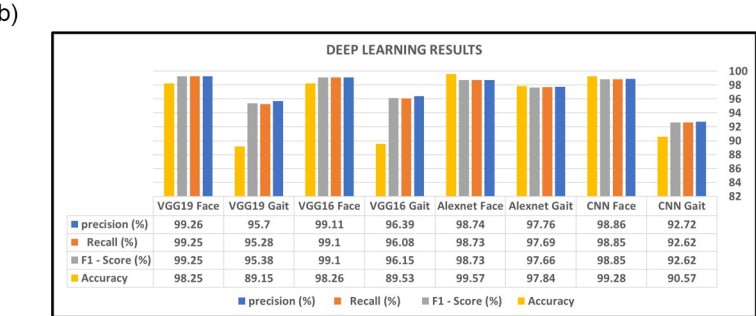

**Fig 7. Machine and deep learning results.** (a) Machine learning results. (b) Deep learning results.

(a)

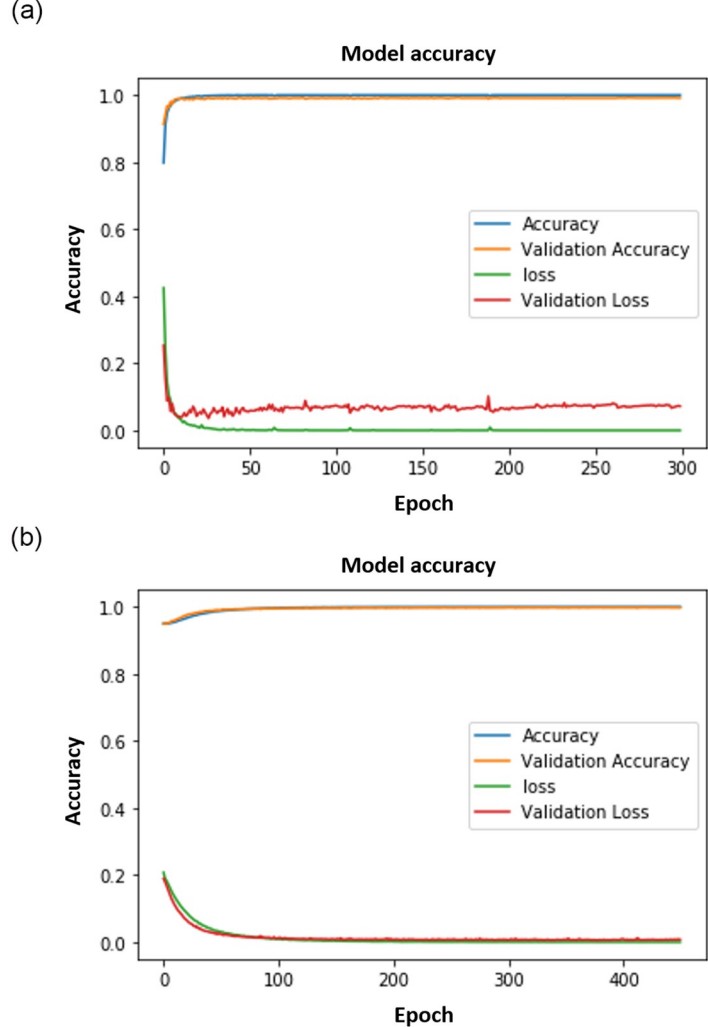

(b)

**Fig 8. AlexNet results.** (a) AlexNet Face. (b) AlexNet Gait.

patterns and poses unique challenges. The ability of VGG19 to learn hierarchical features may improve gait feature extraction, but the simplicity of traditional CNNs could offer advantages in terms of computational efficiency. Detailed examination of the results, considering metrics such as precision, recall, and computational resources, is essential to draw meaningful conclusions and guide the selection of the most suitable model for specific face detection and gait recognition applications.

## 4.4 Performance comparison with other state-of-art approaches

While face detection has been a widely studied area in computer vision, gait detection has emerged as another important aspect of human detection. Gait detection involves identifying and recognizing an individual based on their walking pattern.

In terms of performance comparison, both face and gait detection have been extensively researched, and several state-of-the-art approaches have been proposed. Deep learning-based approaches have shown impressive performance in both tasks.

(a)

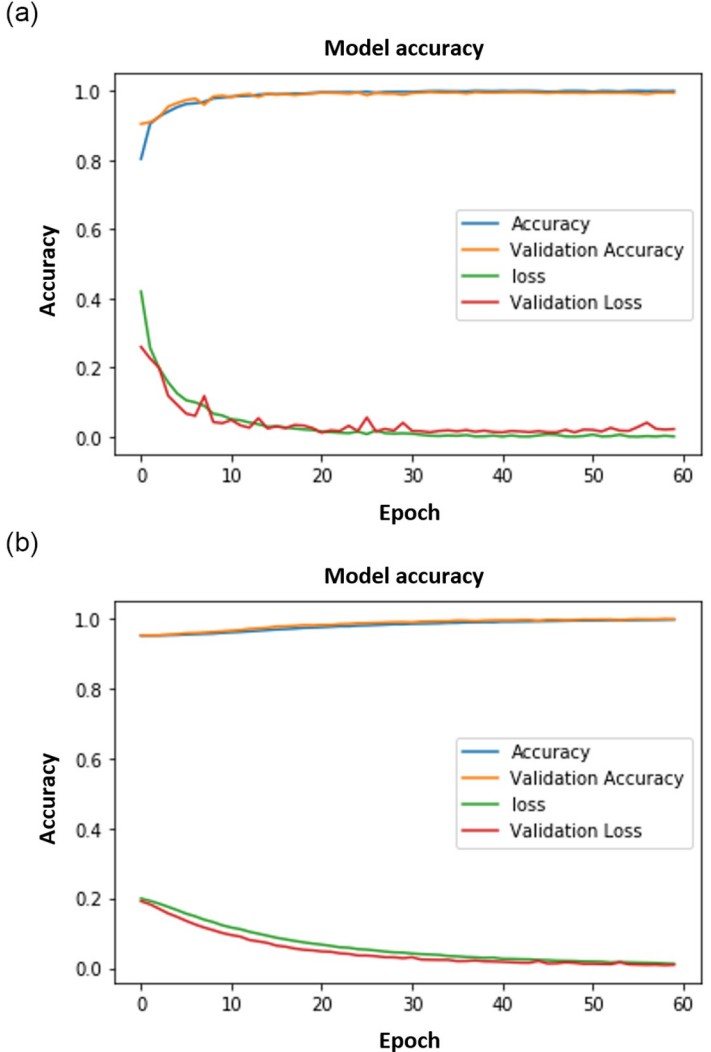

(b)

**Fig 9. VGG16 results.** (a) VGG16 Face. (b) VGG16 Gait.

For face detection, CNN-based architectures such as VGG16, Inception, and AlexNet have been used to extract facial features, which are then fed into different classification models to detect faces. Deep learning-based methods have proven to be more accurate than traditional methods such as Viola-Jones and HOG-based methods.

Likewise, for gait detection, CNN-based architectures are used for extracting and analysing gait features from videos, and then different classification models are used to identify individuals. The most common approaches include using 3D Convolutional Neural Networks (CNN) or their variants (such as spatiotemporal networks) to extract features from multiple frames of a video, which could improve the accuracy of gait recognition.

Generally, the performance of a face or gait detection system is measured in terms of accuracy, recall, precision, and f1-score such as in Tables 2 and 3. The accuracy of a face or gait detection system is measured by the number of correctly identified faces or gaits. The robustness of a face or gait detection system is measured by its ability to detect faces or gaits in different environments and conditions.

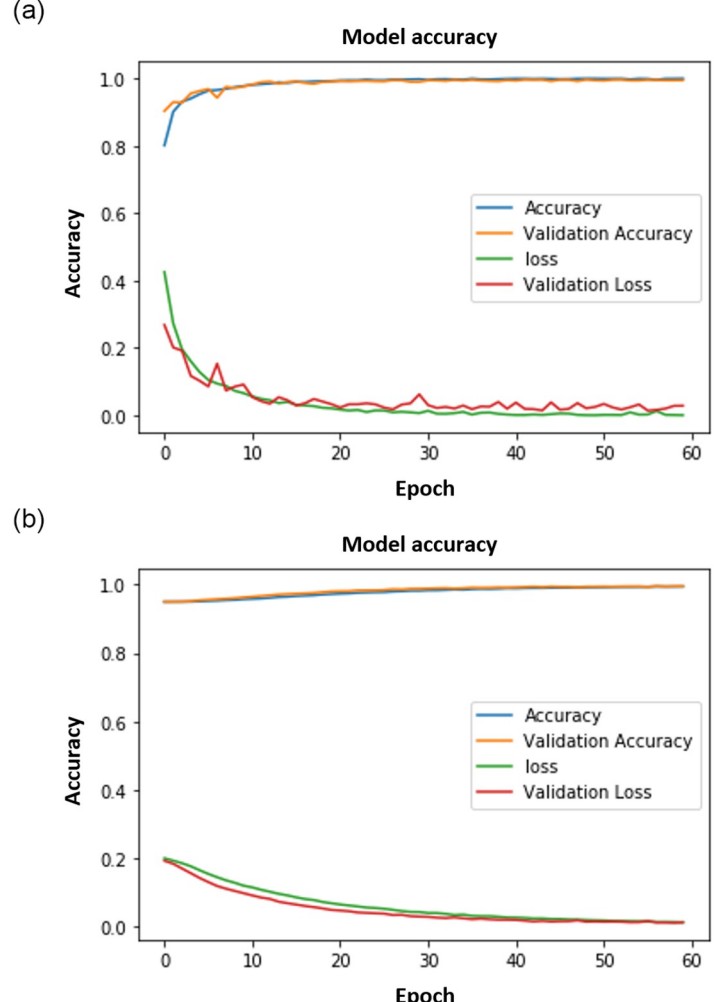

**Fig 10. VGG19 results.** (a) VGG19 Face. (b) VGG19 Gait.

The results depicted in Tables 4 and 5 serve to reinforce the accuracy of our human face and gait detection systems compared to contemporary approaches. In face detection, our system achieved 99.57% accuracy, the second-best position. This success extended across three distinct categories: with mask, without mask, and incorrectly worn mask, showcasing the robustness of our approach. In gait detection, our deep learning-based system claimed the second position with 97.9% accuracy and an F1-score of 97.66%, surpassing a conventional gait detection system that managed only 97.2% accuracy. Our gait detection method can detect 20 individuals across diverse view angles (parallel, 45, and 90 degrees), a significant improvement over other methods. This can be attributed to the strategic use of high hyperparameter values in our CNN architectures, emphasizing the critical role of optimizing model parameters for enhanced performance in human face and gait detection tasks. Our hybrid human face and gait detection systems have proven to be not only more accurate but also more robust, making them well-suited for applications such as security and surveillance, where precision is very important. The simulation results of our hybrid human recognition framework, which includes CNN, VGG16, VGG19, AlexNet, and machine learning algorithms, show a stable and

(a)

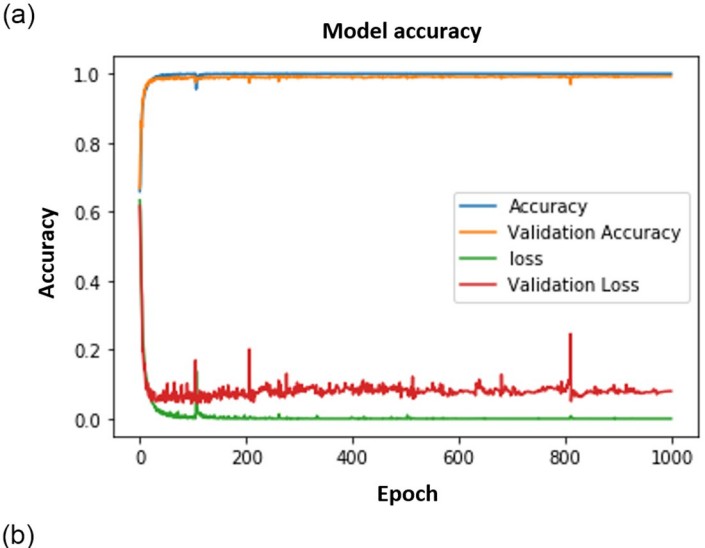

(b)

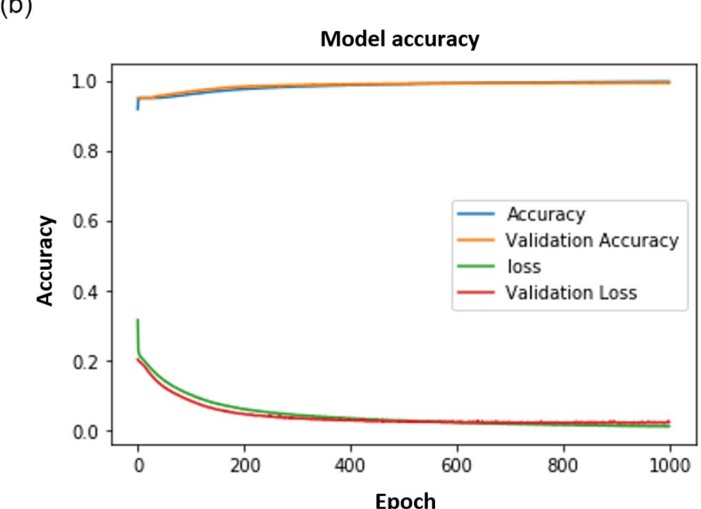

**Fig 11. CNN results.** (a) CNN Face. (b) CNN Gait.

adaptable system with improved performance in various scenarios. Through our simulations, we carefully evaluated the framework's performance in complex scenarios, offering a thorough assessment of its advantages.

In face recognition, our system performed exceptionally well in situations when there were non-direct gaze, low lighting, and people with different types of face coverings. Notably, the

**Table 3. Performance assessments of different detection deep learning architectures for human masked face dataset.**

| Class | AlexNet | | | VGG16 | | | VGG19 | | | CNN | | |
|---|---|---|---|---|---|---|---|---|---|---|---|---|
| | Precision (%) | Recall (%) | F1-Score (%) | Precision (%) | Recall (%) | f1-Score (%) | Precision (%) | Recall (%) | F1-Score (%) | Precision (%) | Recall (%) | F1-Score (%) |
| Mask Incorrect | 98.68 | 100 | 99.33 | 98.68 | 100 | 99.33 | 98.68 | 100 | 99.33 | 98.31 | 100 | 99.14 |
| With Mask | 99.54 | 96.88 | 98.19 | 99.32 | 98.44 | 98.88 | 100 | 98.44 | 99.21 | 99.68 | 97.26 | 98.46 |
| Without Mask | 98.02 | 99.33 | 98.67 | 99.32 | 98.88 | 99.1 | 99.11 | 99.33 | 99.22 | 98.59 | 99.29 | 98.94 |

**Table 4. Comparison between our approach and previous research results based on Human face detection.**

| Ref. | Year | Type of Detection | Classification model | Software library | Best Accuracy (%) |
|---|---|---|---|---|---|
| [18] | 2021 | mask/no mask | MobileNetV2 | TensorFlow, OpenCV | 92 |
| [45] | 2021 | mask/no mask | AdaBoost | Python | 90.9 |
| [46] | 2021 | mask/no mask | MovileNetV2 | PyTorch, OpenCV | 79.2 |
| [47] | 2021 | mask/no mask | VGG-16 CNN | TensorFlow, OpenCV | 96 |
| [48] | 2021 | mask/no mask | CNN | TensorFlow, OpenCV, Caffe | - |
| [49] | 2021 | mask/no mask | CNN | TensorFlow, Keras | 99 |
| [50] | 2020 | mask/no mask | CNN+SVM | | - |
| [51] | 2021 | mask/no mask | ResNet50+SVM | | 99.5 |
| [52] | 2020 | mask/no mask/nose out | MobileNet | OpenCV, Keras | 90 |
| [53] | 2021 | mask/no mask/nose out | InceptionV2 | OpenCV, Matlab | 91.1 |
| [54] | 2021 | mask/no mask/nose out | VGG-16, MobileNetV2,InceptionV3, ResNet50 | Keras | **99.8** |
| [55] | 2020 | mask/no mask/nose out | SRCNet | Matlab | 98.7 |
| [56] | 2023 | mask/no mask | Customized CNN | TensorFlow, OpenCV | 97.5 |
| Our approach | 2024 | mask/no mask/Incorrect Mask | VGG-16, VGG-19, AlexNet, CNN | TensorFlow, Keras | **99.57** |
| Our approach (CNN+ML) | 2024 | mask/no mask/Incorrect Mask | VGG-16, VGG-19, AlexNet, CNN | TensorFlow, Keras | **99.29** |

use of VGG16 and VGG19 assisted in capturing delicate facial traits as shown in Fig 12, while the versatility of CNNs ensured effective processing, even in resource-constrained contexts. Furthermore, the integration of emotion and identity identification based on facial expressions highlighted the framework's innovative capacity to identify subtle human characteristics, distinguishing it from conventional recognition techniques.

**Table 5. Comparison between our approach and previous research results based on Human gait detection.**

| Ref. | Year | Input Type | Classes | Model | Result | Measure |
|---|---|---|---|---|---|---|
| [57] | 2019 | GEI | Three angles | GAN | 82% | Recognition results |
| [58] | 2023 | Images Frame | Three angles | Features fusion | 99.5% | f1-score results |
| [59] | 2024 | Sequence of gait images | 20 persons | Convolutional neural network (CNN) with silhouette gait image | 92% | f1-score results |
| [59] | 2024 | Sequence of gait images | 20 persons | Convolutional neural network (CNN) with Gait Energy Image (GEI) | 91% | f1-score results |
| [59] | 2024 | Sequence of gait images | 20 persons | Lightweight Convolutional neural network (LWCNN) with Gait Energy Image (GEI) | 89% | f1-score results |
| [59] | 2024 | Sequence of gait images | 20 persons | Convolutional neural network (CNN) with Enhanced Gait Energy Image (EGEI) | 90% | f1-score results |
| [59] | 2024 | Sequence of gait images | 20 persons | Lightweight Convolutional neural network (LWCNN) with Enhanced Gait Energy Image (EGEI) | 93% | f1-score results |
| [59] | 2024 | Sequence of gait images | 20 persons | network (CNN) with Hamming distance correlated Enhanced Gait Energy Image (EGEI) and Modified Contrast Limited Adaptive Histogram Equalization (MCLAHE) | 94% | f1-score results |
| [59] | 2024 | Sequence of gait images | 20 persons | GRLNet | 96% | f1-score results |
| Our approach | 2024 | Images Frame | 20 persons | Hybrid face and gait | 97.66% | f1-score results |
| Our approach (CNN+ML) | 2024 | Images Frame | 20 persons | Hybrid face and gait | 98.97% | f1-score results |

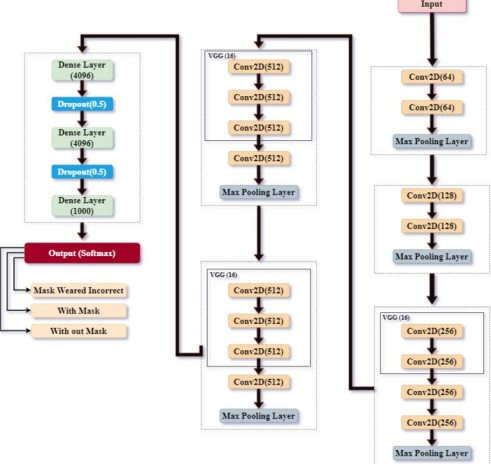

**Fig 12. VGG16 and 19 architectures.**

One important goal was identifying whether individuals were wearing masks, and our framework performed exceptionally well in this regard by utilizing AlexNet's advantages for effective feature extraction. In addition, acknowledging those who wore hats and spectacles or otherwise covered a large section of their faces demonstrated the model's adaptability. By adding CNNs, the difficulties caused by different accessories were overcome and fast and precise detection was made possible.

Using CNNs and components derived from VGG architectures, in the scenario of identifying individuals based on walking patterns, our system performed quite effectively. The extensive review included a variety of viewpoints, guaranteeing that the model could be used in actual surveillance situations. Also included in the simulations was a detailed examination of the various contributions made by CNN, VGG16, VGG19, AlexNet, and VGG16 to the wider framework. The comprehensive discussion outlined each architecture's advantages and explained how to combine them to get the best results possible on challenges involving human recognition. One unique feature of our method is its ability to smoothly combine these various models while preserving computational efficiency.

The enhanced simulation results highlight the flexibility, accuracy, and efficiency of our hybrid human recognition framework in complex circumstances. Our technique is an excellent solution in the human recognition technology field because of its unique identity detection, thorough case evaluations, and the use of different architectures. As further research is conducted, we hope to further improve and broaden the framework's functionality, securing its place as a top tool for a variety of human recognition applications.

## 4.5 Comparison of Pre-trained CNN and From scratch CNN

CNN has demonstrated outstanding performance in developing deep-learning models for computer vision tasks. However, it can be time-consuming and expensive to train CNNs from scratch but does provide more control and flexibility. Using a pre-trained CNN model as a feature extractor, like the VGG (Visual Geometry Group) model, is one technique to circumvent this issue.

We can feed the input images of Mask and Human Gait datasets through the network and extract the activations from one of the intermediate layers to extract deep features using a pre-

**Table 6. Results of scratch and transfer learning.**

| Dataset | Approach of training | Accuracy | Precision | Recall | f1-score |
|---|---|---|---|---|---|
| Human Face dataset https://www.kaggle.com/datasets/vijaykumar1799/face-mask-detection | Transfer learning | 98.28 | 98.30 | 98.28 | 98.28 |
| Human Face dataset https://www.kaggle.com/datasets/vijaykumar1799/face-mask-detection | From scratch | 98.99 | 98.99 | 98.99 | 98.99 |
| Human Gait dataset http://www.cbsr.ia.ac.cn/english/Gait%20Databases.asp | Transfer learning | 94.65 | 94.92 | 94.65 | 94.77 |
| Human Gait dataset http://www.cbsr.ia.ac.cn/english/Gait%20Databases.asp | From scratch | 98.72 | 98.80 | 98.72 | 98.76 |

trained VGG model. These activations, known as "deep features" or "bottleneck features," record highly abstract representations of the input images. Once the deep features have been extracted, they can be fed into another classifier, like a Random Forest. A combination of multiple decision trees is used in the Random Forest combination learning method. It has gained popularity for classification because of its popularity and durability in handling high-dimensional feature spaces.

To use Random Forest as a classifier, the model must be trained on the extracted deep features. Providing training data that has been labelled involves using deep features as input and associated class labels as outputs. The Random Forest approach will develop a mapping between the deep features and the corresponding classes, and then build a decision function using the obtained knowledge.

After training, the Random Forest model is used to predict the class labels of images that have not yet been seen by feeding it the deep features obtained from the images. The learnt decision function will be applied to these deep features by the Random Forest, which will then produce the expected class label. Using pre-trained CNNs like VGG as feature extractors, followed by training a Random Forest classifier on the extracted deep features, is known as transfer learning. In our case, these from-scratch models lead to better accuracy compared to training pre-trained models. Table 6. displays the results.

## 4.6 Extedended experiments

In this research, we introduced a novel approach for detecting masked faces and recognizing gait with a high degree of accuracy. Our methodology leverages a stacked scratch-built learning convolutional neural network (CNN) model, demonstrating its efficacy through extensive comparative studies and evaluations against state-of-the-art classification models. We developed a custom CNN model designed specifically for the detection of masked faces and gait recognition. This scratch-built architecture was tailored to the intricacies of the datasets, ensuring optimal feature extraction. To enhance feature learning, we employed ensemble deep learning by utilizing CNNs as base learners. This approach allowed us to extract diverse and complementary features from the input images, contributing to a more robust and accurate model. We further enriched our methodology by integrating features extracted from four well-established deep learning classifiers: CNN, VGG16, AlexNet, and ResNet34. These features were combined and fed into machine learning classifiers, including KNN, RF, DT, and XGBoost. The experimental results demonstrated exceptional accuracy, with our approach achieving rates between 98.51% to 98.95% as shown in Table 7. This indicates the reliability and effectiveness of our model in detecting masked faces and recognizing gait. Our approach outperformed contemporary masked face and gait recognition detectors when using combined deep learning architecture features in machine learning classifiers. The results showcase the superiority of our integrated model in terms of accuracy and robustness. We achieved success rates ranging between 98.97% to 99.29% based on recall, precision, and F1-Score as shown in Table 8. These

**Table 7. Performance assessments of different detection deep learning architectures for human masked face and gait datasets.**

| Dataset | CNN | | | VGG16 | | | AlexNet | | | ResNet34 | | |
|---|---|---|---|---|---|---|---|---|---|---|---|---|
| | Precision (%) | Recall (%) | F1-Score (%) | Precision (%) | Recall (%) | f1-Score (%) | Precision (%) | Recall (%) | F1-Score (%) | Precision (%) | Recall (%) | F1-Score (%) |
| Masked face | 98.00 | 97.98 | 97.97 | 98.99 | 98.99 | 98.99 | 98.92 | 98.92 | 98.92 | 98.96 | 98.95 | 98.95 |
| Gait Human | 96.02 | 95.99 | 95.99 | 98.24 | 98.21 | 98.21 | 96.84 | 96.75 | 96.77 | 98.52 | 98.51 | 98.51 |

**Table 8. Performance assessments of different detection machine learning algorithms using deep learning features for human masked face and gait datasets.**

| Dataset | KNN | | | RF | | | DT | | | XGboost | | |
|---|---|---|---|---|---|---|---|---|---|---|---|---|
| | Precision (%) | Recall (%) | F1-Score (%) | Precision (%) | Recall (%) | f1-Score (%) | Precision (%) | Recall (%) | F1-Score (%) | Precision (%) | Recall (%) | F1-Score (%) |
| Masked face | 99.29 | 99.29 | 99.29 | 99.29 | 98.85 | 99.07 | 98.83 | 98.82 | 98.82 | 98.00 | 97.98 | 97.97 |
| Gait Human | 98.97 | 98.97 | 98.97 | 99.42 | 98.02 | 98.71 | 97.08 | 96.71 | 96.78 | 96.12 | 95.93 | 96.01 |

metrics emphasize the comprehensive nature of our model, which excels in both sensitivity and precision, crucial for applications in human detection.

Our research presents a powerful and accurate solution for masked face and gait recognition through the integration of a stacked scratch-built CNN model and ensemble deep learning. The combination of deep learning architectures and machine learning classifiers resulted in outstanding accuracy, surpassing contemporary detectors. These findings underscore the potential of our approach for practical applications, emphasizing its robustness and effectiveness in addressing challenges related to human detection in diverse scenarios.

## 4.7 Execution time performances

In our study, we conducted a comprehensive evaluation of the execution time and scalability of our approach, comparing various deep learning architectures and machine learning classifiers on two distinct datasets: Masked Face and Gait Human. The results are presented in Tables 9 and 10. According to deep learning architectures execution time as shown in Table 9,

**Table 9. Execution time performances of deep learning architectures.**

| Dataset | CNN | VGG16 | AlexNet | ResNet34 |
|---|---|---|---|---|
| | (seconds) | (seconds) | (seconds) | (seconds) |
| Masked face | 904.5 | 1,508.4 | 603.3 | 1,057.2 |
| Gait Human | 3,007.2 | 6,015 | 1,503.9 | 4,210.2 |

**Table 10. Execution time performances of machine learning classifiers.**

| Dataset | KNN | RF | DT | XGboost |
|---|---|---|---|---|
| | (seconds) | (seconds) | (seconds) | (seconds) |
| Masked face | 1.004 | 2.008 | 1.003 | 1.006 |
| Gait Human | 3.008 | 6.015 | 2.005 | 5.012 |

AlexNet exhibited the fastest execution time on the Masked Face dataset, completing the task in approximately 10 minutes. Traditional CNN architecture followed, taking 15 minutes. ResNet34 and VGG16 had progressively longer execution times due to their deeper architectures.

For the Masked Face dataset, KNN, Decision Tree (DT), and XGBoost exhibited similar and relatively short execution times of around 1 second as shown in Table 10. Random Forest (RF) took slightly longer, approximately 2 seconds. On the Gait Human dataset, DT outperformed other classifiers with a quick execution time of 2 seconds. KNN and XGBoost followed, taking approximately 3 and 5 seconds, respectively. RF had the longest execution time, around 6 seconds, reflecting the larger size of the Gait Human dataset.

Deep learning architectures, and machine learning classifiers significantly influences execution times. AlexNet proved to be the fastest among deep learning architectures for the Masked Face dataset, while machine learning classifiers exhibited varying speeds depending on the nature of the dataset and algorithm. These findings provide valuable insights for selecting the most suitable models based on execution time requirements in applications such as facial recognition and gait analysis.

## 5 Discussion and limitations

A potential approach for recognizing biometric data is the Hybrid Face and Gait Recognition Framework, which uses deep neural networks such as CNN, VGG16, VGG19, AlexNet, and machine learning. However, to thoroughly grasp the framework's capabilities and opportunities for growth, the constraints must be explored and discussed.

According to computational intensity, deep neural network architectures add a significant computational load, especially when architectures like VGG16 and VGG19 are used. The viability of the framework in situations with limited resources or on-edge devices may be limited by the significant computer resources required for training and implementing these models.

The diversity and quality of the training data have a significant impact on the framework's performance. A dataset not fully representative of real-world scenarios could introduce biases and make it more difficult for the model to generalize, which could impact the model's dependability in real-world applications. Deep neural networks frequently have interpretability issues, which causes difficulty in understanding the reasoning behind recognition judgments. The absence of explainability in delicate situations, such as security applications, can make people wonder if the technology is reliable.

The computational needs of complicated neural network topologies may make it difficult to achieve real-time processing, a critical requirement in applications such as surveillance. The usefulness of the framework for time-sensitive scenarios may be affected by its limited capacity to respond quickly.

The framework may find it difficult to adjust to changes in environmental factors such as weather and lighting. Maintaining consistent performance in a variety of scenarios is difficult but essential to the system's usefulness in the actual world.

The use of facial recognition technology presents issues with permission, privacy, and abuse. To guarantee their ethical implementation, a balance must be drawn between protecting the rights of individuals and developing technological improvements.

The efficacy of the recognition framework can vary across different demographic groups, resulting in potential disparities in performance. Ensuring equitable performance across diverse populations is critical to avoid biases and discriminatory outcomes.

Although deep neural networks such as VGG16 and VGG19 have significantly contributed to the framework, the input from shallower networks such as AlexNet and conventional CNNs

is much less. Future research should ascertain how the synergies between these systems can be maximized.

The Hybrid Human Recognition Framework has significant potential. However, cautious evaluation of these constraints is required for its continual improvement and appropriate implementation. To maintain ethical standards in the use of facial recognition software and assure the framework's broad applicability, future research should address issues such as computing efficiency, interpretability, and ethical considerations.

## 6 Threat of validity

Masked face recognition has received little recent research. A lack of common masked facial recognition datasets is one reason for this. However, several diverse and large-scale unmasked facial recognition datasets include the facial images of many thousands of people.

There is no widely used, standardised masked face recognition dataset. Unlike an unconvincing face, a masked face dataset involves photographs of various persons wearing masks, as well as information on their skin tone, race, gender, light, and gait.

There is a gap in the research because there is currently no relevant dataset for masked faces and walking. Applications for masked facial recognition and face mask detection cannot be evaluated using a single dataset. Masks cover a large portion of the face, including the nose and lips, which could affect facial recognition performance, which is necessary to enable contactless operation. As a result, many characteristics cannot be retrieved.

### 6.1 Conclusion

In this study, we focus on identifying a person from a photo or video by comparing facial features, walking, and identifying them using a person's biometrics, and reducing the weaknesses of person recognition techniques, such as when a person is not looking directly at the camera, the lighting is poor, or the person is covering their face well.

To improve the results, we therefore suggest a deep learning and machine learning approach and an analysis and comparison of the results from deep learning and machine learning algorithms. Our approach is divided into four parts. Finding a person with or without a mask is the first step; finding people who cover big portions of their faces is the second step; and finding the person if they walk is the third step.

Our hybrid human recognition framework marks a substantial leap forward in the domain of human recognition systems by using a combination of machine learning and deep neural networks, including popular architectures such as VGG16, VGG19, AlexNet, and CNN. The main advantages of our proposed approach are its resilience and flexibility, which combine the best features of these many designs to produce a strong and flexible framework. Our methodology is unique because it strategically integrates these well-known models, taking advantage of their special qualities to improve identification accuracy and efficiency in a range of difficult situations.

One of the more noteworthy advancements is the capability of our framework to effortlessly blend the effectiveness and simplicity of standard CNNs with the hierarchical feature learning of deep neural networks such as VGG16, VGG19, and AlexNet. This combination helps the system to operate more efficiently in identifying people in situations such as indirect gaze, dim lighting, and full-face coverings. In addition, our system performs exceptionally well when integrating sophisticated aspects like identification and emotion detection from facial expressions, offering a sophisticated and all-encompassing method of human recognition.

Our approach is significant as it simultaneously takes into consideration several factors in the recognition process. Our system covers a range of real-world difficulties, from efficiently

detecting people wearing or not wearing masks to identifying those who are concealing major areas of their faces, such as those wearing hats, glasses, and masks. Furthermore, our ability to recognize people based on their distinct gait patterns offers an additional degree of protection, demonstrating our methodology's adaptability.

By contrasting well-known architectures such as CNN, VGG16, VGG19, AlexNet, etc., our study advances the field by stating the advantages and disadvantages of each and clarifying how best they can be combined for maximum efficacy. In the field of human recognition technology, Our approach is innovative because it can seamlessly combine various models while maintaining cost and computational effectiveness. We hope that our framework will continue to be improved and expanded upon in future, enhancing its position as a state-of-the-art, flexible, and important tool for a broad range of human recognition applications.

Consequently, this approach has been developed to help successfully find a person who poses no risk. This paper makes the following contributions.

- We built a full study to detect people wearing masks, people not wearing masks, and people who cover a large part of their face by some other means, such as wearing a hat, glasses, and mask, and finding them by walking.

- We compared deep learning and machine learning and ascertained which is better to find people.

- We developed our approach to identify people at the lowest possible cost. We believe the idea can be applied in the real world and continue to explore how the framework can be generalized, given its high commercial significance.

In the future, we plan to extend our work to create human masked faces and gait detection for the same people and further improve masked face detection and walking using unsupervised and supervised learning algorithms.

## Author Contributions

**Conceptualization:** Abdullah M. Sheneamer.

**Data curation:** Abdullah M. Sheneamer, Malik H. Halawi.

**Formal analysis:** Abdullah M. Sheneamer, Malik H. Halawi.

**Funding acquisition:** Abdullah M. Sheneamer.

**Investigation:** Abdullah M. Sheneamer, Malik H. Halawi, Meshari H. Al-Qahtani.

**Methodology:** Abdullah M. Sheneamer, Malik H. Halawi, Meshari H. Al-Qahtani.

**Project administration:** Abdullah M. Sheneamer.

**Software:** Abdullah M. Sheneamer, Malik H. Halawi, Meshari H. Al-Qahtani.

**Supervision:** Abdullah M. Sheneamer.

**Validation:** Abdullah M. Sheneamer.

**Visualization:** Abdullah M. Sheneamer.

**Writing – original draft:** Abdullah M. Sheneamer, Malik H. Halawi, Meshari H. Al-Qahtani.

**Writing – review & editing:** Abdullah M. Sheneamer, Malik H. Halawi, Meshari H. Al-Qahtani.

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
