## [Decision Letter · Decision Letter 0]

5 Dec 2023

PONE-D-23-35791A Hybrid Human Recognition Framework Using Machine Learning and Deep Neural NetworksPLOS ONE

Dear Dr. Sheneamer,

Thank you for submitting your manuscript to PLOS ONE. After careful consideration, we feel that it has merit but does not fully meet PLOS ONE’s publication criteria as it currently stands. Therefore, we invite you to submit a revised version of the manuscript that addresses the points raised during the review process.

We look forward to receiving your revised manuscript.

Kind regards,

Priyadarsan Parida, Ph.D.

Academic Editor

PLOS ONE

Journal Requirements:

Reviewers' comments:

Reviewer's Responses to Questions

**Comments to the Author**

1. Is the manuscript technically sound, and do the data support the conclusions?

Reviewer #1: Yes

Reviewer #2: Partly

2. Has the statistical analysis been performed appropriately and rigorously? 

Reviewer #1: Yes

Reviewer #2: Yes

3. Have the authors made all data underlying the findings in their manuscript fully available?

Reviewer #1: Yes

Reviewer #2: Yes

4. Is the manuscript presented in an intelligible fashion and written in standard English?

Reviewer #1: Yes

Reviewer #2: Yes

5. Review Comments to the Author

Reviewer #1: Comments to the Author:

The general idea of the paper seems to be good. The following questions or suggestions are expected to be discussed.

1. The authors need to provide proper references for the InceptionV3 model and Simulated Masked Face dataset in subsection 2.2.

2. The acronym “SMFD” should be placed after the Simulated Masked Face dataset. In subsection 2.2, it is wrongly placed after the InceptionV3 model.

3. In subsection 2.2, the authors used the acronym PCA for primary part analysis, which should be PPA. Commonly, the acronym PCA stands for principal component analysis.

4. In section 2.4, the authors have written “Johansson made the first attempt to prove this in 1970 when he discovered that….”. Please provide the reference.

5. In section 2.5, Authors discussed about “Automated Person Identification System Using Walking Pattern Biometrics. However, related works in Gait Recognition are still not enough. Some related works as provided below should be discussed.

i. Rani V, Kumar M (2023) Human gait recognition: A systematic review. Multimedia Tools Appl 1–35

ii. R.P. Sankara, G. Sahu, P. Parida, A contemporary survey on human gait recognition. J. Inf. Assur. Secur. 15(3), 94–106 (2020)

iii. Mogan, J.N., Lee, C.P., Lim, K.M.: Advances in vision-based gait recognition: From handcrafted to deep learning. Sensors 22(15), 5682(2022)

6. How the hyperparameter values (Table 1) have been chosen for training the proposed model?

7. The obtained results shown in Figure 11 & 12 need considerably more explanation.

8. In Table 4, the CNN model [29] has achieved 60% accuracy, but the proposed CNN model has achieved 99.57%. What are the factors/improvements that helped the proposed CNN model to achieve 99.57% accuracy?

9. In Table 5, The proposed approach is compared with only two previous research results. The author should compare with other recent approaches and give more technical comments.

10. References missing for datasets presented in Table 6.

11. Simulation Results shall be enriched. More detailed scenarios shall be added.

12. The conclusion is justified, but it could be extended, highlighting the advantages of the proposed method and specify exactly what is new?

13. I see few grammatical mistakes and unnecessary capitalization of words in the middle of sentences. Hence, the manuscript needs proper proofreading.

Reviewer #2: 1. Keywords section is missing.

2. Describe dataset features in more details and its total size and size of (train/test).

3. Pseudocode / Flowchart and algorithm steps need to be inserted.

4. Time spent need to be measured in the experimental results.

5. Limitation and Discussion Sections need to be inserted.

6. All metrics need to be calculated in the experimental results (Loss).

7. The parameters used for the analysis must be provided in table

8. The authors need to make a clear proofread to avoid grammatical mistakes and typo errors.

9. Add future work in last section (conclusion) (if any)

10. To improve the Related Work and Introduction sections authors are recommended to review this highly related research work paper:

a) Innovative Hybrid Approach for Masked Face Recognition Using Pretrained Mask Detection and Segmentation, Robust PCA, and KNN Classifier

b) An accurate system for face detection and recognition

c) A Robust and Efficient System to Detect Human Faces Based on Facial Features

d) Face recognition based on Grey Wolf Optimization for feature selection

e) An Effective Hybrid Method for Face Detection

6. PLOS authors have the option to publish the peer review history of their article (what does this mean?). If published, this will include your full peer review and any attached files.

Reviewer #1: No

Reviewer #2: **Yes: **Tarek Abd El-Hafeez

---

## [Author Response · Author response to Decision Letter 0]

9 Feb 2024

We have looked at issues/errors raised by the reviewers and have made appropriate corrections.

Details of changes we have made are described in the attached file. The entire editorial decision letter written by the Editor-in-Chief also is included.

Thanks,

Authors

---

## [Decision Letter · Decision Letter 1]

4 Mar 2024

A Hybrid Human Recognition Framework Using Machine Learning and Deep Neural Networks

PONE-D-23-35791R1

Dear Dr. Sheneamer,

We’re pleased to inform you that your manuscript has been judged scientifically suitable for publication and will be formally accepted for publication once it meets all outstanding technical requirements.

Kind regards,

Priyadarsan Parida, Ph.D.

Academic Editor

PLOS ONE

Additional Editor Comments (optional):

Reviewers' comments:

Reviewer's Responses to Questions

**Comments to the Author**

1. If the authors have adequately addressed your comments raised in a previous round of review and you feel that this manuscript is now acceptable for publication, you may indicate that here to bypass the “Comments to the Author” section, enter your conflict of interest statement in the “Confidential to Editor” section, and submit your "Accept" recommendation.

Reviewer #1: All comments have been addressed

Reviewer #2: All comments have been addressed

2. Is the manuscript technically sound, and do the data support the conclusions?

Reviewer #1: Yes

Reviewer #2: Partly

3. Has the statistical analysis been performed appropriately and rigorously? 

Reviewer #1: Yes

Reviewer #2: Yes

4. Have the authors made all data underlying the findings in their manuscript fully available?

Reviewer #1: Yes

Reviewer #2: (No Response)

5. Is the manuscript presented in an intelligible fashion and written in standard English?

Reviewer #1: Yes

Reviewer #2: Yes

6. Review Comments to the Author

Reviewer #1: Reviewers Comments:

In this paper “A Hybrid Human Recognition Framework Using Machine Learning and Deep Neural Networks” is proposed and the advantages of the proposed method are demonstrated by some examples. Authors response is appropriate and the submission is worthy of publication.

Reviewer #2: "All feedback provided by reviewers has been thoroughly addressed, ensuring that all concerns and suggestions have been carefully considered and implemented. Therefore, I confidently recommend moving forward with the publication of the manuscript."

7. PLOS authors have the option to publish the peer review history of their article (what does this mean?). If published, this will include your full peer review and any attached files.

Reviewer #1: No

Reviewer #2: **Yes: **Tarek Abd El-Hafeez

---

## [Editor Report · Acceptance letter]

1 Apr 2024

PONE-D-23-35791R1 

PLOS ONE

Dear Dr. Sheneamer, 

I'm pleased to inform you that your manuscript has been deemed suitable for publication in PLOS ONE. Congratulations! Your manuscript is now being handed over to our production team.

Kind regards, 

on behalf of

Dr. Priyadarsan Parida 

Academic Editor

PLOS ONE